# Lifelong Fitness in Ambulatory Children and Adolescents with Cerebral Palsy I: Key Ingredients for Bone and Muscle Health

**DOI:** 10.3390/bs13070539

**Published:** 2023-06-28

**Authors:** Noelle G. Moreau, Kathleen M. Friel, Robyn K. Fuchs, Sudarshan Dayanidhi, Theresa Sukal-Moulton, Marybeth Grant-Beuttler, Mark D. Peterson, Richard D. Stevenson, Susan V. Duff

**Affiliations:** 1Department of Physical Therapy, School of Allied Health Professions, Louisiana State University Health Sciences Center, New Orleans, LA 70112, USA; 2Burke Neurological Institute, Weill Cornell Medicine, White Plains, NY 10605, USA; kaf3001@med.cornell.edu; 3Division of Biomedical Science, College of Osteopathic Medicine, Marian University, Indianapolis, IN 46222, USA; rfuchs@marian.edu; 4Shirley Ryan Ability Lab, Chicago, IL 60611, USA; sdayanidhi@sralab.org; 5Department of Physical Therapy & Human Movement Sciences, Northwestern University, Chicago, IL 60611, USA; theresa-moulton@northwestern.edu; 6Department of Physical Therapy, Oregon Institute of Technology, Klamath Falls, OR 97601, USA; marybeth.grantbeuttler@oit.edu; 7Department of Physical Medicine and Rehabilitation, Michigan Medicine, University of Michigan, Ann Arbor, MI 48109, USA; mdpeterz@med.umich.edu; 8Division of Neurodevelopmental and Behavioral Pediatrics, Department of Pediatrics, School of Medicine, University of Virginia, Charlottesville, VA 22903, USA; rds8z@uvahealth.org; 9Department of Physical Therapy, Crean College of Health and Behavioral Sciences, Chapman University, Irvine, CA 92618, USA; duff@chapman.edu

**Keywords:** muscle strength, muscle power, resistance training, bone health, lifelong fitness

## Abstract

Physical activity of a sufficient amount and intensity is essential to health and the prevention of a sedentary lifestyle in all children as they transition into adolescence and adulthood. While fostering a fit lifestyle in all children can be challenging, it may be even more so for those with cerebral palsy (CP). Evidence suggests that bone and muscle health can improve with targeted exercise programs for children with CP. Yet, it is not clear how musculoskeletal improvements are sustained into adulthood. In this perspective, we introduce key ingredients and guidelines to promote bone and muscle health in ambulatory children with CP (GMFCS I–III), which could lay the foundation for sustained fitness and musculoskeletal health as they transition from childhood to adolescence and adulthood. First, one must consider crucial characteristics of the skeletal and muscular systems as well as key factors to augment bone and muscle integrity. Second, to build a better foundation, we must consider critical time periods and essential ingredients for programming. Finally, to foster the sustainability of a fit lifestyle, we must encourage commitment and self-initiated action while ensuring the attainment of skill acquisition and function. Thus, the overall objective of this perspective paper is to guide exercise programming and community implementation to truly alter lifelong fitness in persons with CP.

## 1. Current Physical Activity Guidelines for Children and Adolescents

The World Health Organization (WHO) [1] and the American College of Sports Medicine (ACSM) [2] recommend that children and adolescents achieve a minimum of 60 min of physical activity per day at a moderate to vigorous intensity to maintain health. Both groups recommend involvement in vigorous-intense aerobic activities three times per week to support the integrity of the developing musculoskeletal and cardiopulmonary systems. Despite these published guidelines and benefits, many children and adolescents who are typically developing (TD) fall below recommended levels [3], and those with neurodevelopmental conditions such as cerebral palsy (CP) are at even greater risk for achieving insufficient physical activity [4,5]. We believe that these data, demonstrating deficient physical activity, should serve as a call to action for fostering lifelong fitness in all children, particularly adolescents at risk for the secondary effects of a sedentary lifestyle, such as those with CP.

Physical activity is linked to quality of life and happiness [6,7]; thus, strategies to enhance adherence to programming should be holistic in nature and salient to the individual performer [8]. Empowering children and adolescents to be self-directed in their choice of activities is a powerful link to sustaining change and preventing a sedentary lifestyle. Long-term adherence to physical activity can also strengthen the musculoskeletal (MSK) system and sustain the shorter-term gains in MSK health seen during focused fitness programs. The positive impact of interventions targeting bone and muscle health demonstrated in children with CP further amplifies the need for opportunities to participate in activities that benefit these body systems. The purpose of this perspective paper is to identify key ingredients for interventions targeting the MSK system and provide guidelines for promoting bone and muscle health that are essential to programming for achieving the goal of lifelong fitness for persons with CP across the Gross Motor Function Classification Scale (GMFCS), particularly those who are ambulatory (GMFCS levels I–III). Further, embedding a framework of lifestyle intervention into programming could help empower children to be self-directed, fostering motivation and habit formation essential to sustaining change in the MSK system and preventing a sedentary lifestyle as they move into adolescence and adulthood [9].

## 2. Musculoskeletal (MSK) System

Adults living with CP and other pediatric-onset disabilities that target the MSK system have a significantly higher prevalence of common psychological, cardiometabolic, and musculoskeletal morbidity and multimorbidity as compared to adults without CP [10,11,12]. Changes in the muscular system contribute to the onset of sedentary behavior, which is pervasive in children and adolescents with CP—even those at higher levels on the GMFCS [4,13,14]. Therefore, it is important to consider the characteristics of the MSK system and related functions in persons with CP during the design of programs to augment the integrity of this system and improve function in this group at risk.

The health of the MSK system is influenced by insufficient physical activity, especially during the transition from adolescence to adulthood. The risk of progressive, age-and activity-related declines in MSK health is even greater for persons with CP [15,16,17]. Changes in bone and muscle integrity and associated function are well documented in those with CP, particularly during development and into adulthood [18,19,20,21,22,23,24]. A reduction in muscle growth has been found as early as 15 months of age [25]. As children move into adolescence and adulthood, suboptimal nutritional and mechanical factors can negatively influence the integrity of the MSK system, thereby reducing tolerance to physical activity [24,26].

### 2.1. Skeletal System

Bone mass, cross-sectional bone size, and bone strength increase rapidly in childhood and peak in adolescence for TD persons, as shown in Figure 1. Individuals that attain optimal peak bone mass have a reduced risk of developing osteoporosis and sustaining a low-trauma fracture [27]. The development and maintenance of bone mass require adequate mechanical loading to stimulate structural and mineral adaptations. Engaging in moderate- to high-level physical activity during childhood and adolescence contributes to optimizing peak bone mass and the ability to sustain it into adulthood [28,29].

**Figure 1 behavsci-13-00539-f001:**
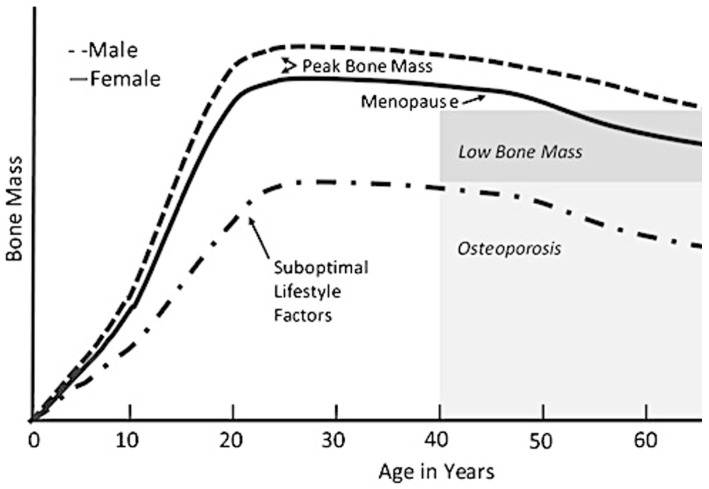
Bone mass across the lifespan with optimal and suboptimal lifestyle choices. Reproduced with permission from Weaver et al. [30].

Gunter and colleagues [31] stress that activities that have the greatest effect on developing bone mass are those with a high magnitude of force applied at a rapid rate. Figure 2 depicts the ground reaction force (GRF) in units of bodyweight and the time to peak force for low- and high-impact activities performed by a representative 10-year-old girl. The authors report that activities with the most osteogenic potential have GRFs greater than 3.5 times body weight (per leg), with peak force occurring in less than 0.1 s. Jumping activities from a 100 cm box had GRFs of 8.5 times body weight and were found to improve hip bone mass in children [32,33].

**Figure 2 behavsci-13-00539-f002:**
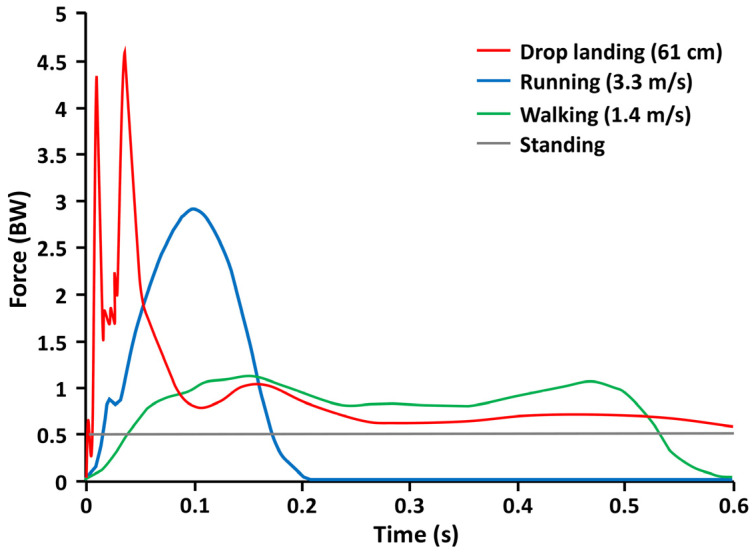
Ground reaction forces and rates of loading for **(1)** drop landings from a 61-cm box with a peak force of 4.5 BW/leg (red); **(2)** running at a speed of 3.3 m/s resulting in a peak force of 3 BW/leg (blue); **(3)** w at a speed of 1.4 m/s resulting in a peak force of 1 BW/leg (green); and **(4)** quiet standing resulting in a static load of 0.5 BW/leg (gray). This illustration of unpublished data by Jeremy J. Bauer was adapted and used with his permission [34].

Inactivity in childhood has been linked to an increased incidence of osteoporosis and osteoarthritis later in life [31,35,36]. The National Osteoporosis Foundation advocates for high physical activity and the intake of calcium (~1000–1300 mg/d) and vitamin D (600 IU/d) in childhood and adolescence to enhance bone health [30,37]. A decrease in activity and mobility in persons with CP reduces mechanical loading vital for bone integrity [24,38]. Yet, it is important to consider whether children and adolescents with CP who have alterations in skeletal alignment and strength can tolerate sufficient impact loading on an ongoing basis to improve and sustain bone health. It is also unknown whether a lower-impact activity could provide adequate osteogenic potential to augment bone integrity in persons with CP. Therefore, it is important to consider individualized exercise prescriptions that safely expand impact loading through physical activity in persons with CP across the GMFCS levels I–III. Further, prescriptions that safely target all GMFCS levels to increase bone mass and sustain bone health into adulthood must be investigated.

Adolescents who are TD accumulate 25% of their adult bone mass during the two years after peak height velocity, occurring from about 11.8 to 13.4 years of age [36]. Bone minerals accrue in those with CP during adolescence and young adulthood but remain significantly below those of the general population [39]. Deficiencies in bone mineral accrual are multifactorial and include inadequate nutrition and suboptimal physical activity [40]. By the time those with CP enter adulthood, their mean bone density may be greater than 2 standard deviations below the mean for age and sex, depending on GMFCS levels [41]. The impact on bone mass is also dependent on the severity of CP, with children that have limited mechanical loading having lower bone health compared to TD children and those with a lower GMFCS level [39,42,43]. Moderate to severe impairment in persons with CP, along with inactivity [5], contribute to the higher prevalence of fractures [36,44,45]. To have a positive influence on long-term bone health requires the targeted intervention at critical time points during development, with optimal gains in bone mass and size prior to late childhood and early adolescence to take advantage of the accrual associated with pubertal growth [46].

Skeletal maturation and integrity vary within and across GMFCS levels. Henderson et al. [47] found that in children and adolescents 2.6 to 21 years of age with moderate to severe CP, skeletal age closely approximated chronological age. However, deficiencies in skeletal maturation were found to be associated with delays in height, low lumbar bone density, and poor nutrition status. Clinically, these could be used as markers to evaluate the risk of low bone mass in children with CP across all GMFCS levels. Chen et al. [48] examined skeletal integrity in children with CP with a range in severity and found lower limb bone density to be correlated with increasing GMFCS level. Vertebral fracture risk is increased in children with GMFCS levels IV/V, with those children at GMFCS levels I–III having a similar incidence of fracture as TD children [49,50]. Paying close attention to the integrity of the skeletal system before and during intense physical activity and impact loading is vital to preventing injury during any training program, regardless of GMFCS level.

### 2.2. Muscular System

The properties of healthy muscle, including architecture, elasticity, connective tissue, and sarcomeres, typically adapt to functional demands and use [51]. As children grow, muscle strength and motor skills progressively increase [52]. Sufficient muscle development, strength, power, and adaptability are essential before engaging in demanding physical activities and specific motor skills (e.g., sports). Thus, those who lack sufficient strength and skill may be less competent and confident in their performance abilities, leading to less frequent engagement in physical activity. To ensure safety, muscle integrity must be considered before attempting to augment muscle function with demanding physical activity.

Atypical muscle growth contributes to altered physiology and MSK integrity [18,25,53,54,55]. Muscle volume and passive mechanical properties of those with CP have been reported to differ from those who are TD as early as 1–3 years of age [25,54]. Multiple factors contribute to variable growth, passive stiffness, and the onset of muscle contracture. At a cellular level, muscle myofiber areas in persons with CP do not appear to develop at the same rate as in those who are TD (Figure 3) [53]. Muscle sarcomeres typically increase in series in response to chronic stretching, including during postnatal development [56,57,58]. Yet, muscle contractures in persons with CP demonstrate overstretched muscles and fewer serial sarcomeres [59,60], suggesting an inability to add sarcomeres during growth. In CP, muscle contractures and a reduction in muscle stem cells (satellite cells) with abnormalities in their function are reported, along with an increase in the extracellular matrix [53,61,62,63,64]. Importantly, this might not be true for all muscles or at younger ages [65]. The muscle substrate of those with CP has also been found to have a reduction in mitochondrial function and content, which is important for energy production [66,67]. Interestingly, the onset of fat infiltration can alter the ability of affected muscles to generate adequate force [68,69,70]. These multiple alterations in muscle for those with CP are important to consider during training since sufficient loading is essential for muscle plasticity and muscle performance.

At the macroscopic level, measures of muscle size, such as muscle volume, cross-sectional area, and muscle thickness, are significantly decreased in children and adolescents with CP as compared to TD peers [18,19,23,71,72]. Muscle size is a strong predictor of force output, but force-generating capacity is also negatively impacted by changes in both passive and active mechanical properties that were discussed previously. Moreover, diminished muscle size is a significant contributor to decreased muscular strength and is strongly influenced by mobility levels as measured by the GMFCS [73]. While deficits in voluntary activation are present in CP and do play a role, these deficits appear to be more pronounced in the plantarflexors as compared to the quadriceps [74].

During typical development, muscle strength increases until it peaks between 20 and 30 years of age, then slowly declines. In persons with CP, muscle strength and power increase at a lower rate, peak at an earlier age, and are hypothesized to contribute to a faster decline with age than those without CP [75] (Figure 4). This loss of strength and muscular reserve is believed to result in early and rapid age-related decreases in function. This time period also corresponds to a period of ambulatory decline as individuals with CP transition into adolescence and adulthood [76,77,78,79].

In addition to muscular strength, Moreau [80] stresses the importance of muscle power for the performance of functional activities. Muscle power and a sufficient rate of force development (RFD) are essential for motor transitions during gait, stairclimbing, and other functional tasks such as transfers and are also important for reducing fall risk. Moreau reported that muscle power is more significantly impaired than strength in persons with CP compared to those who are TD [81,82]. In addition, while knee extensor muscle power increases linearly with age in persons who are TD, the rate of increase is lower for those with CP as they move from childhood to young adulthood [83] (Figure 5). Investigations into the neuromuscular adaptations that occur in response to specific activity and exercise have been examined in CP [81,83,84], yet further work is needed across the GMFCS spectrum.

In a recent review article, Faigenbaum et al. [85] remarks on the decline in muscular fitness in 7-year-old TD children and adolescents. This decline and the incidence of sub-optimal physical activity levels place children and adolescents at risk for injury and adverse health conditions. Faigenbaum et al. [85] advocate for early resistance training and postulate that strength gains after training are often related to neural and muscular adaptations. Given the high degree of neuroplasticity in pre-adolescence, the development of muscular strength, power, and motor skill performance should be emphasized in childhood in preparation for projected gains in adolescence [86,87]. In addition to the neuromuscular and musculoskeletal benefits of resistance training, there is longstanding literature documenting the robust association between strength and cardiometabolic health among children and adolescents [88,89].

## 3. Building a Better Foundation for MSK Health in Children with CP

Despite the risk of progressive declines in fitness, health, and function in persons with CP, the optimal age and key program ingredients essential to ensuring a long-lasting level of fitness, health, and function are largely unknown. However, current and related data can be leveraged to begin program design and monitoring. Creating a strong physical health foundation that supports optimal MSK development should include critical considerations, as reviewed below.

### 3.1. Critical Time Periods

Given the number of factors contributing to atypical MSK growth and development in persons with CP [47], it is vital to intervene early enough and at an adequate intensity and dose to have a long-lasting effect on muscle and bone integrity. Targeted intervention at critical time points during development is needed, particularly in childhood and early adolescence, to optimize peak bone mineral accrual associated with pubertal growth. Ideally, this would occur prior to peak height velocity in order to significantly influence muscle and bone health [36,39], allowing for successful participation in fitness and leisure activities of interest. This is also the optimal period for building a strong muscular reserve, as muscular strength peaks in early adulthood before slowly declining. In children with CP, the rate of increase in muscular strength during growth is less than TD, thus reaching peak strength earlier and potentially contributing to an earlier decline in muscular strength and function over the lifespan (Figure 4). Therefore, intervening during childhood and pre-adolescence is necessary to potentially restore the natural history of strength optimization early in life and attenuate the decline that occurs in adulthood. In Figure 4, we propose an altered trajectory of age-related decline secondary to targeted interventions delivered during the pre-adolescent critical period.

### 3.2. Key Ingredients

Programs that aim to improve and sustain MSK integrity require specific physical activities to target key components of morphology and functional capacity. Recommended guidelines for exercise and physical activity in persons with CP have been proposed [45,80]. In addition, we suggest that programs include activities aimed at improving bone and muscle integrity as well as cardiorespiratory reserve tailored to GMFCS level with consideration of tolerance as well as level of interest.

#### 3.2.1. Targeted Musculoskeletal Intervention

Augmenting Bone Health. Sufficient physical activity that provides muscular stimulus and impact forces that target osteogenesis could prevent osteoporosis and reduce the risk of fracture. Yet, to ensure that the skeletal system of a person with CP can tolerate force-related and impact loading activities, pre-testing of bone density is required. Bone mineral content (BMC; g) and areal bone mineral density (BMD); g/cm^2^) are two clinical measures of bone health that can be assessed with dual X-ray absorptiometry (DXA). Knowing the patient’s bone health will ensure that precautions are taken to ensure a child at risk can begin to safely engage in physical activity. For children with low bone mass based on calculated Z-scores, it is still safe to exercise. For example, a child with a higher GMFCS level may have low bone mass, but if placed in a harness or assisted with the exercises, the individual can still benefit from the exercise while minimizing fall risk. Ultimately, fractures are primarily caused by an impact force from a fall or landing that exceeds the failure properties of the bone tissue [90]. Given the limited data in children on the risk of fracture during exercise and what the skeleton can tolerate, we can glean some insight from exercise interventions in osteoporotic women who are performing high-intensity exercise. In these studies, women with osteoporosis were able to handle high impact loads and did not fracture [91].

Based on published and unpublished data [32,34] by Bauer et al. and Gunter et al. [31], GRFs per body weight (BW) for activities performed by a TD child are about 1.0 times BW for walking, 2.9 times BW for running, and 4.6 times BW for drop landing (see Figure 2). The GRFs per BW for activities performed by a child with CP are less widely known and may be strongly influenced by select impairments such as ligamentous laxity, joint deformity, body malalignment, inadequate passive and active range-of-motion, and insufficient eccentric muscular control. Quick, high-load tasks that a child with CP at GMFCS levels I–III may tolerate could include jumping rope, hopscotch, or jump downs off a bench. Individuals with mobility at GMFCS level III may require the use of a harness or walker for support and balance during loading tasks. Tasks with a low impact load, such as jumps on a mini-trampoline, may be safer, but they may not provide sufficient GRFs per BW to influence changes in bone mass and structure. It may be best to have a child participate in circuit training, which may include intermittent impact loading, allowing for periodic monitoring of safety and tolerance. For example, the sequence of a course could be: (1) hopscotch; (2) jump rope; (3) crawling through tubes; and (4) jump downs. Further study is needed to ascertain the types of loads safely tolerated by persons with CP at all levels of the GMFCS.

Enhancing Muscle Performance. As reviewed, there are important age-related changes in the muscles of persons with CP that differ from those who are TD. Despite these differences, muscle hypertrophy, force production, and power can increase in children and adolescents with CP who undergo targeted training at a sufficient dosage [81,92]. Because muscle architecture can differentially adapt in response to different types of resistance training [81], the type of training and dosing essential to altering muscle function must be incorporated into programming (Section 3.2.2. Dosing Parameters).

Strength training is recommended to build a strong muscular foundation, promote muscle hypertrophy, and provide a synergistic stimulus for bone health. Yet, the effects of traditional strength training have not been shown to carry over to activities such as gait and functional mobility in those with CP [93,94]. Power training, which involves training at moderate to high loads at a higher concentric velocity of movement, is recommended for better carryover to gait and functional activities. Targeted high-velocity training may not only increase muscle power but also induce muscle architectural adaptations, such as an increase in fascicle length and cross-sectional area, and promote a right-ward shift of the torque-angle curve, increasing torque production at higher velocity [81,83].Traditional resistive training equipment (i.e., free weights and isotonic machines) is readily available in most gyms and clinics and can easily be used for strength and power training. Basic bodyweight exercises can also be used, especially in very young children, and can be progressed to free weights, machines, or other loaded exercises. Other modifications for children at GMFCS level III may include the use of support walkers for balance and to encourage hands-free positioning in standing while promoting weightbearing through the lower limbs. Regardless of GMFCS level, the advantages of using weight machines are that the child can be supported and single or multiple joints can be isolated while preventing or discouraging compensatory patterns and unwanted movements. For example, an inclined leg press can be used to train and target multiple lower extremity muscle groups while safely supporting the trunk and body [95,96]. In this supported position, the muscles can be loaded in a safe manner to a greater extent than if the same movement was attempted as an upright standing squat.While there are selected types of equipment that provide precise measures of velocity and force, such as isokinetic equipment, these are not necessary for resistance training purposes. Power training can be feasibly conducted on most equipment by moving a constant load while decreasing the amount of time allowed to produce the concentric contraction (i.e., increasing the velocity). For example, a power leg press can be performed on an inclined leg press machine and would train and target the hip and knee extensors and ankle plantarflexors with a single exercise [95]. Typical verbal instructions include “Push, pull, or press as fast as possible” and “lower slow and controlled”, referring to the concentric and eccentric portions of the motion, respectively. Once a sufficient velocity is reached, the load should be increased. Instrumented versions can also be used to reliably measure power output while performing a power leg press [95]. Another example of equipment that can be used for power training is flywheel ergometers. The equipment can be in the mode of a bike, rower, or ski machine that couples resistance from the device with the speed of active motion while providing digital power output. In a randomized crossover study by Moreau and colleagues [97] in persons with CP, 7 to 24 years of age, power output in the upper extremities significantly increased after 15 training sessions using an upper extremity flywheel ergometer (Concept2 SkiErg™, Morrisville, VT, USA). Use of the device at home or in school strengthened adherence.Community-based training alternatives are also important for promoting mobility-based participation. RaceRunning (or Frame Running), which uses a three-wheeled running frame, is an example of how children within GMFCS levels I–IV can successfully engage in community-based sports programming if provided with adaptation [98]. Further, muscle hypertrophy and an increase in cardiorespiratory endurance were observed after a 12-week program across a wide age range (9 to 29 years) and mobility levels (GMFCS I–IV) [99]. Training alternatives to improve muscle and bone health while fostering engagement should continue to expand, allowing greater access to this type of programming in various settings.Despite the success of resistance training programs for persons with CP, there are some risks of pain and injury. However, no serious adverse events have been reported for resistance training interventions in children and adolescents with CP. A few studies have reported mild adverse events, such as joint or muscle soreness [100,101]. It is highly recommended that those participating in any resistive or power training program be supervised and monitored closely by a trained professional [80]. Safety and tolerance are key factors for all programs to augment muscle integrity and function.

#### 3.2.2. Dosing Parameters

Exercise prescription includes the parameters of frequency, intensity, volume, duration, and velocity. Frequency considers the number of training sessions per week. Volume refers to the number of sets and repetitions within one session. Intensity indicates the relative load used and is often defined as a percentage of the one-repetition maximum (1RM) or percentage of body weight [80,96]. The duration is the full length of the training program. Velocity refers to the rate and direction in which an exercise is performed. Exercise guidelines differ for bone and muscle, yet both should be strongly considered when designing programs.

Dosing for Bone. Dosing parameters used to guide interventions to improve bone health are often based on guidelines to increase peak bone mass and prevent osteoporosis [27,102], which may be an important consideration for those with CP given the risk factors. General guidelines for TD children have been advocated by Gunter et al. [31]. These guidelines have been framed within the dosing parameters of frequency and volume (Table 1). The authors propose that children engage in 40–60 min of daily weight-bearing activity to target hip structure and strength [103]. Based on their own findings, they recommend 10–15 min of jumping 3 times per week to augment bone mass and structure [33,104]. This frequency and volume equate to 100 jumps from a two-foot height with GRFs at least 3.5 BW and higher). Table 1 includes examples of bone-building exercises that could be performed in children across all GMFCS levels, with associated ideas for how to modify activities. Since tolerance to skeletal loading varies across the GMFCS spectrum, methods to augment bone health in persons with CP must consider the individual integrity of the skeletal system and monitor safety and tolerance throughout the training program.

Dosing for Muscle. Recommended optimal dosing guidelines for progressive resistance training have been assembled as shown in Table 2, specific to muscle strengthening vs. power training [80]. Novice lifters should begin training at a lower intensity (percentage of 1RM) as described in Moreau [80] in more detail and then progress up to the optimal dosage provided in Table 2 in order to maximize muscle plasticity. For example, a novice may begin power training at 40% of 1RM and focus on form and speed, then progress to a higher percentage of 1RM after successful completion of the target reps at the higher concentric velocity. Of note is that intensity, volume, and speed differ between the two training paradigms. It is important that a 1RM test be performed to adequately dose the intensity of the intervention and the progression of the intensity throughout the intervention period. The safety, feasibility, and protocol for performing a 1RM in youth with CP have recently been published by Pontiff and Moreau [96]. Although a multiple repetition maximum test may be used to predict 1RM values, the prediction is less accurate for repetition ranges greater than 10 [105]. Regardless of what muscle performance parameter is being targeted, the recommended frequency for resistance training is 2 to 3 times per week on nonconsecutive days for a duration of 8 to 20 weeks (refer to Moreau, 2020 for more details) [80]. A recent review article by Moreau and Lieber [83] on resistance training interventions for youth with CP showed that if the optimal dosing guidelines were adhered to, then muscle plasticity was observed at the macroscopic structural level (i.e., increases in cross-sectional area, muscle thickness, volume, or fascicle lengths).

### 3.3. Maximizing Engagement and Addressing Barriers

Youth are significantly influenced by environmental factors, including relationships with peers and mentors. Sport-based youth development is a strategy that aims to promote healthy behaviors in conjunction with social confidence [106] through athletic games, team building, and emotional learning opportunities. These principles can be adjusted to sports or activities of interest to children of different ability levels. It is important to offer various types of activities that may be of interest to lay a strong foundation of MSK health that could be maintained over a lifetime.

Fortunately, the variety of physical activities available to children with CP is expanding. Grant–Beuttler and colleagues [107] delivered a Balanced Families Dance Program for children with CP and other disabling conditions. They have shown that a child with diplegic CP at GMFCS Level II was able to improve his scores on the GMFM through ongoing engagement in the Dance Program. Sukal–Moulton and colleagues [98] similarly showed an improvement in self-efficacy and self-perception in an adolescent with diplegia following participation in a running program using a running frame for aerobic training, including engagement in a community race. At the conclusion of training, the participant stated, “we are a family of runners now” in reference to joining her parents and younger brother in running for fitness; this was followed by a study where group level differences were found [98]. The inclusion of psychosocial features into any program designed to increase physical activity while augmenting bone and muscle health may significantly contribute to immediate and long-term sustainability and enrich quality of life [108].

Despite the fact that physical activity and exercise are essential to enhance and maintain bone and muscle health, barriers to engagement in fitness activities for children include disinterest, transportation, cost, and time [109]. There are additional barriers to these that significantly impact participation in fitness centers and sports for persons with CP and other disabling conditions. In a study by Wright et al. [110], the key barriers identified by youth with disabilities were the lack of accessible and inclusive opportunities. Even if fitness centers are considered inclusive, Nikolajsen et al. [111] reported that patrons still encounter issues of ableism and disablism from staff and members. *Albeism* places value on self-sufficiency, autonomy, and independence, which can lead to the exclusion of people with disabilities where the diversity of physical form is not represented. *Disablism* refers to the psychosocial oppression that persons with disabilities may experience directly due to negative interactions from staff or club members and indirectly due to structural barriers. Nikolajsen et al. [112] found that while able-bodied members at inclusive fitness centers held the desire for persons with disabilities to feel welcome, the notions of direct and indirect psychosocial disablism did have an influence. These societal constructs and barriers are also seen in youth sports programs, where participation is too often limited to those with a specific set of physical skills and the ability to navigate a variety of environments. Specific barriers for people with disabilities include not being physically capable but also societal attitudes, frustration, and loss of confidence. Bringing fitness and social opportunities to the home setting via tele-rehabilitation may address transportation needs and inclusion but may not offer the maximal engagement and positive encouragement that have been shown to be vital to increasing the confidence and skills of participants [109]. These offerings, as well as other options for virtual exercise, may reduce the burden by eliminating the need for travel. Virtual exercise can be effective and engaging, though it is important to determine which specific types and doses of exercises are best suited to a particular individual at their current level of abilities. Programs focused on inclusion must consider many elements of access, including activities, physical spaces, understanding participant needs, and intentional actions to shift attitudes and expectations of anyone interacting with the program.

## 4. Sustaining Gains

Introducing sustainable physical activity options at an early age that could augment MSK health is essential for persons with CP. To ensure that fitness is maintained into adolescence and adulthood, programs must be engaging and fit into one’s interests and lifestyle [9]. We propose that a comprehensive, individualized multi-modal exercise program introduced in pre-adolescence may provide the optimal stimulus to enhance the integrity of multiple systems, prevent the acquisition of a sedentary lifestyle, and contribute to positive gains in self-efficacy.

## 5. Conclusions

The recent initiative to improve physical activity in children and adolescents with CP requires attention to key ingredients and dosing parameters to augment muscle and bone health. The dosing guidelines and recommendations put forth here can guide designers in planning multi-modal exercise programs that include impact loading and resistance training for individuals with CP at a sufficient but safe level. Based on research evidence, we recommend building a strong foundation of MSK health in pre-adolescence through participation in individualized exercise activities that are engaging, enjoyable, and promote skill acquisition. Sustaining fitness requires the integration of healthy habits into one’s lifestyle. With a strong MSK foundation merged into a healthy lifestyle, lifelong fitness can be achievable for persons with CP.

## Figures and Tables

**Figure 3 behavsci-13-00539-f003:**
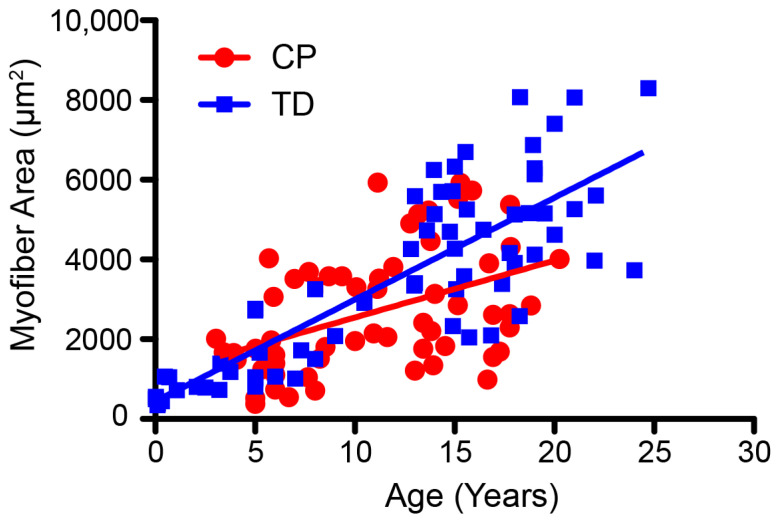
Change in mean myofiber area for age in TD children (squares, *n* = 67) and those with CP (circles, *n* = 58). Data were extracted based on published cross-sectional studies.

**Figure 4 behavsci-13-00539-f004:**
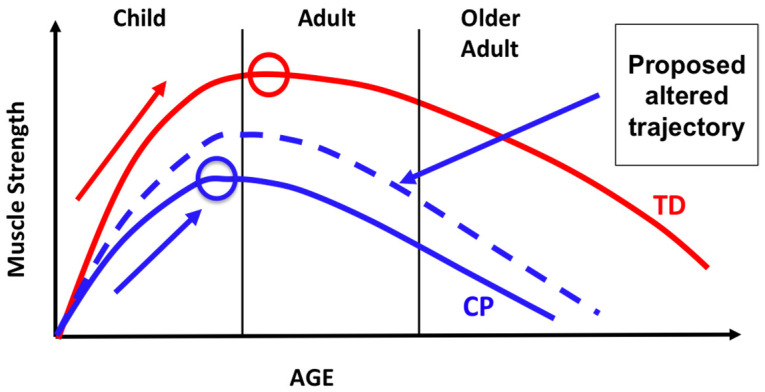
Age-related changes in muscle strength for TD individuals (red line) and those with CP (blue line). Circles signify peak muscle strength for TD (red) and CP (blue). The dashed blue line denotes a proposed altered trajectory of age-related decline with targeted intervention during the pre-adolescent critical period, including lifelong changes in fitness, health, and function.

**Figure 5 behavsci-13-00539-f005:**
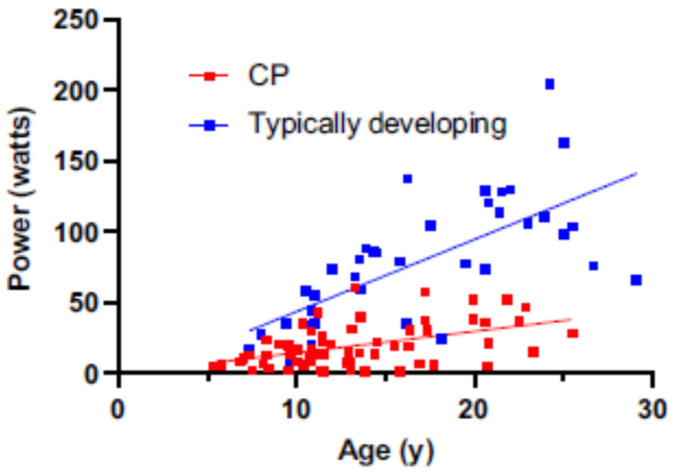
Relationship between knee extensor power and age in a TD (blue circles) cohort (*n* = 42; ages 7–29 y) and in a cohort with CP (red circles) (*n* = 66; ages 5–25 y). Data compiled from unpublished and published cross-sectional data. Reproduced with permission from Moreau and Lieber, 2022.

**Table 1 behavsci-13-00539-t001:** Recommended optimal dosing guidelines for bone health. Examples of different types of exercises that dynamically load the skeleton to stimulate osteogenesis. The duration required to stimulate a bone response is longer.

Intensity *	Volume	Skeletal Site ^†^	Speed	Duration ^#^	Rest
Body weight	100 jumps of boxes of varying heights up to 24 inches.	Hip, spine	Controlled, landing with both feet	3–6 mths	15–30 sec between jumps
Body weight	100 jump circuit (hopscotch, jump ups, skips, side jumps) from floor height.	Hip, spine	Controlled landing with both feet	3–6 mths	15–30 sec between jumps
Body weight	Jump roping, 5–10 min (~50 jumps/min)	Hip, spine	Controlled, landing with both feet	3–6 mths	

* These could be performed using a harness system for children with higher GMFCS levels to provide added support. Some studies have had participants add a weighted vest for added loading, with weights between 1 and 3 kg. ^†^ Only the bones that are mechanically loaded will respond. ^#^ Evaluate changes in bone by DXA after 6–12 months.

**Table 2 behavsci-13-00539-t002:** Optimal dosing parameters for strength vs. power training.

Parameter	Intensity	Volume	Speed	Frequency	Duration	Rest
Muscle strength	70% to 85% of 1RM	3 sets of 6 to 10 repetitions	Slow and controlled to moderate (concentric and eccentric)	2–3 × per week (nonconsecutive days)	8–20 weeks	1–2 min. between sets; 48 h between sessions
Muscle power	60% to 80% of 1RM	3–6 sets of 1 to 6 repetitions	Concentric: As fast as possible Eccentric: Slow and controlled over 2–3 s	2–3 × per week (nonconsecutive days)	8–20 weeks	1–2 min. between sets; 48 h between sessions

From Moreau (2020); 1RM: one repetition maximum.

## Data Availability

Not applicable.

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
