# Peer review of "Lifelong Fitness in Ambulatory Children and Adolescents with Cerebral Palsy I: Key Ingredients for Bone and Muscle Health"

_behavsci, 2023, doi:10.3390/bs13070539_

Round 1

Reviewer 1 Report

This opinion or perspective piece discusses the literature on physical fitness, bone and muscle health for individuals with CP. It is well written, and structured to address this issue. The authors draw on evidence related to typical development and applies this where evidence is lacking for individuals with CP. The authors make a strong case for promoting physical fitness starting in childhood and maintaining through adolescence into adulthood by integrating into appropriate and meaningful lifestyle activities.

My main comment is that this perspective is primarily aimed at GMFCS I and II, with some inclusion of GMFCS III. Although the authors make reference to certain activities (like jumping) that could be done with a harness at higher GMFCS levels, minimal information relevant to GMFCS IV/V is provided. Also, children at GMFCS III may crawl, but <50% use walking as a primary means of mobility prior to school-age. The authors note that muscle changes begin as early as 15 months for children with CP, but the ideas discussed in this article are aimed at older children and pre-adolescence. If the aim is for this article to be primarily aimed at GMFCS I-III as stated on page 2 line 64 (‘particularly for those who are ambulatory’) and page 3 line 116 - then more information and ideas relevant to GMFCS III should be included. In early childhood, for example, support walkers may be used to encourage upright positioning, weightbearing through the lower limbs and hands-free positioning in standing. Rather than using the arms for support, building lower-limb and trunk strength is important so the arms can be used for balance and steering. An article was recently published evaluating the effects of power training for children using hand-held and support walkers (GMFCS III and IV) during adapted sports that may be useful to consider.

I recognize that it would be challenging to add all the information relevant to the entire GMFCS spectrum - and also address the age spectrum as is the intention of this perspective. However, the abstract should be modified to indicate clearly the GMFCS levels addressed - and if the intent is to include individuals who ambulate with aids (GMFCS III) - then this should be addressed more clearly with appropriate interventions beginning prior to 15 months when muscle development starts to diverge from typical development. Lifelong fitness should include the early childhood period also.

My other points are minor:

Page 4 line 141: ‘vertebral fracture risk is increased in children with GMFCS levels >3’. Firstly GMFCS should be roman numerals (III) - but maybe it would be clearer to say ‘GMFCS IV/V’.

Page 9 line 368 there is a typo - GMFCF - instead of GMFCS

Reviewer 2 Report

This Perspective is quite well-written and addresses an important gap in the literature. I have only a couple critiques.

-As authors indicate, goals of physical exercise and potential benefits go beyond the MSK system (including potential benefits in cardiovascular/respiratory health, metabolic status, mental health domains). It is reasonable to focus on MSK-related goals, but the article title should reflect this

-Recommendations in 3.2.2 appear to be based on suggestions from previous references-- is evidence truly adequate to deem these exemplar regimens "optimal"-- particularly, as authors indicate, in populations of individuals with CP with varying functional capabilities and risks of adverse effects?
